# OpenReview forum: "CertainlyUncertain: A Benchmark and Metric for Multimodal Epistemic and Aleatoric Awareness"
_ICLR.cc/2025/Conference — ICLR 2025 Poster_

### Official Review · Reviewer_EAqJ · 2024-10-19

**Soundness:** 3
**Presentation:** 3
**Contribution:** 3
**Rating:** 6
**Confidence:** 4

**Summary:**

This paper tackles the critical challenge of uncertainty in vision-language AI systems, providing a structured taxonomy that distinguishes epistemic (information-based) from aleatoric (inherent unpredictability) uncertainties, with finer subcategories. Leveraging this taxonomy, the authors introduce a novel dataset, CERTAINLY-UNCERTAIN, comprising 178K VQA examples organized as contrastive pairs. These pairs are generated through image inpainting and language model prompting to showcase instances where answers transition between certainty and uncertainty. The paper also proposes a new metric, confidence-weighted accuracy, to evaluate model performance by integrating both accuracy and calibration. Experiments reveal that large vision-language models (LVLMs) demonstrate significant weaknesses in uncertainty awareness, though fine-tuning on the proposed dataset mitigates calibration errors, improves refusal-based benchmarks, and reduces hallucination rates—achieving these without compromising VQA performance. This work underscores the need for uncertainty-aware AI systems to enhance their reliability and usability in real-world applications.

**Strengths:**

* **Well-Designed Benchmark Dataset:** The authors carefully provide a rigorous classification of uncertainties into two high-level types (epistemic and aleatoric) with six finer categories. The development of 178K samples through a synthetic data pipeline and human curation reflects commendable effort.

* **Innovative Metric:** The proposed confidence-weighted accuracy offers a comprehensive evaluation by factoring in both the correctness and confidence of predictions, penalizing overconfident errors while rewarding high-confidence correct answers.

* **Comprehensive Experiments:** The paper explores multiple training paradigms, including fine-tuning, R-tuning, and preference optimization, across a range of benchmarks. Results highlight both the importance of addressing uncertainty and the effectiveness of the proposed dataset.

**Weaknesses:**

* **Limited Model Coverage:** As a benchmark, I think the number of models evaluated is not sufficient. For closed-source models, the author only evaluated GPT-4V. In some of my experiments, I found that Claude 3.5 performed significantly better than GPT-4V for IDK problems. For open-source models, recent advanced models were not tested, such as InternVL2, Qwen2-VL, and LLaVA-Onevision. Expanding model coverage would better illustrate the benchmark's importance and impact.

* **Limited Evaluation:** Most experiments focus on refusal-oriented tasks with fewer hallucination-related benchmarks. Expanding evaluations to include hallucination datasets like HalluBench [1] and SHR [2], along with general benchmarks such as MME and MMBench, would provide a better view. If models merely improve in refusal scenarios but fail on broader tasks, the contribution of the proposed dataset may appear less impactful.

* **Concern of Data Proportion:** As seen in Table 1, the "Extraneous" awareness category has a higher data ratio compared to other categories. A clearer explanation for this design choice would be beneficial.

* **Concern of "Complexity Awareness" in the benchmark:** While the taxonomy is insightful, certain examples under "complexity awareness" seem to fall into a gray area. For instance, in Figure 7, the query regarding the number of balloons forming "30" could reasonably expect an approximate answer rather than an IDK response. Similarly, for the "distance between the two cats," an estimated answer would seem more appropriate in terms of user's expectation.


[1] Zhao, Zhiyuan, et al. "Beyond hallucinations: Enhancing lvlms through hallucination-aware direct preference optimization." *arXiv preprint https://arxiv.org/abs/2311.16839* (2023).

[2] Guan, Tianrui, et al. "HallusionBench: an advanced diagnostic suite for entangled language hallucination and visual illusion in large vision-language models." *Proceedings of the IEEE/CVF Conference on Computer Vision and Pattern Recognition*. 2024.

**Questions:**

All relevant questions are listed in the “Weaknesses” section.

---

> ### Author Response · Authors · 2024-11-21
> **Response to Reviewer EAqJ (1/2)**
>
> Thank you for the constructive feedback. Please find our response below.
>
> > Q1: Limited Model Coverage
>
> Thanks for the suggestion. Due to limited resources during the rebuttal phase, we report additional evaluation results on Claude 3.5 and Qwen2-VL-7B. We will try our best to include the other models suggested by the reviewer to Table 4 in the final version.
>
>
> - **Evaluation of Claude3.5 on CertainlyUncertain subset**
>
>     We report Claude3.5 performance on the same subset with 100 samples per category as the GPT4V evaluation. We added the detailed results to Table 4 and include a brief comparison on **LAVE Accuracy** below. We observe worse performance of Claude 3.5 than GPT4V, with total accuracy even lower than LLaVA-1.6-34B.
>
>
>     | Model | Epistemic | Aleatoric | Total  |
>     | -------- | -------- | -------- | -------- |
>     | GPT4V     | 78.60     | 61.25     | 71.70     |
>     | Claude-3.5     | 72.92     | 53.25     | 65.10     |
>
>     Our brief examination of Claude 3.5 results suggest that Claude 3.5 is overly confident in terms of temporal and ambiguous categories, making its performance on Aleatoric awareness much worse than GPT4V. While sometimes its prediction is a reasonable guess, Claude 3.5 often state it with certainty. Below is an example from the temporal category, where `events in the past or future that cannot be inferred from the image alone with absolute certainty`.
>
>     > Question: Will the gray cat climb onto the ladder in the future?
>     >
>     > Claude 3.5 response: In this image, I can see a gray cat in the foreground, and there appears to be a black cat and a white cat in the background near what looks like a wooden ladder. The gray cat seems to be focused on something (possibly a red string or toy) rather than showing interest in the ladder at the moment. Based on its body language and direction, it doesn't appear that the gray cat is planning to climb the ladder.
>
>     In addition, Claude3.5 may also suffer from the similar generative AI paradox for generating/understanding “uncertain questions”, as we have observed for GPT4V.
>
>
> - **Evaluation of more recent models Qwen2-VL on the full CertainlyUncertain testing set**
>
>     Below we report Confidence-weighted accuracy of  Qwen2-VL-7B and compare against Qwen-VL-Chat and  LLaVA-1.6-7B in Table 4. The detailed results of  Qwen2-VL-7B can be found in the updated Table 4.
>
>     | Model | Total Confidence-weighted accuracy |
>     | -------- | -------- |
>     | Qwen-VL-Chat     | 13.45     |
>     | LLaVA-1.6-7B     | **20.47**     |
>     | Qwen2-VL-7B     | 18.05     |
>
>
>     From the evaluation of Qwen2-VL-7B, we observe that even though this model has significantly advanced on other multimodal benchmarks (*e.g.*, Qwen2-VL-7B achieving 67 avearage score on OpenCompass [1], with + 20 improvement over Qwen-VL-Chat and LLaVA-1.6-7B), the performance of Qwen2-VL-7B on CertainlyUncertain remain similar to or even slightly lower than that of LLaVA-1.6-7B. This result further supports our motivation: (1) existing benchmarks focus mainly on scenarios where clear and definitive answers are available; and (2) current vision-language models and the corresponding training recipe and data  does not typically encourage the models to express uncertainty or acknowledge when they do not know the answer.
>
> With the fast evolution of LLMs and VLMs, we believe that **the true effective and useful benchmarks should not be static**. As the models evolve, it is possible that some types of uncertainty can be greatly resolved than others, similar to evolving models addressing existing benchmarks. We believe our taxonomy and evaluation can provide a more clear measure on the model progress along this direction. Similar to other dynamic benchmarks like WildBench [2], we can sub-select the questions that majority models fail and include them in the test set to increase difficulty in future iterations. We would like to also highlight that **our proposed uncertainty taxonomy provides insightful and useful guidance in explicitly generating questions for different types of uncertainty.**  Hence, we can re-apply our taxonomy and framework to update the benchmark with the more powerful models, and also extend to new categories of multimodal uncertainties as the research along this direction advances.
>
> [1] [OpenCompass: A Universal Evaluation Platform for Foundation Models](https://github.com/open-compass/opencompass)
>
> [2] Lin, Bill Yuchen, Yuntian Deng, Khyathi Chandu, Faeze Brahman, Abhilasha Ravichander, Valentina Pyatkin, Nouha Dziri, Ronan Le Bras, and Yejin Choi. "WILDBENCH: Benchmarking LLMs with Challenging Tasks from Real Users in the Wild." arXiv preprint arXiv:2406.04770 (2024).

---

> ### Author Response · Authors · 2024-11-21
> **Response to Reviewer EAqJ (2/2)**
>
> > Q2: Limited Evaluation
>
> We follow the reviewer's suggestion to add additional benchmarking results on SHR and MME for the LLaVA-1.5-7B-LORA model variants below.
>
>
> - **Results on SHR**
>
>     We follow [the original implementation of SHR evaluation](), and report the following metrics: Mean Hallucination Ratio, Hallucination Sentence Ratio (hal_sents_ratio), Hallucination Words Ratio (hal_words_ratio), number of hallucination sentences per image (# hal_sents_per_image) and number of hallucination words per image (# hal_words_per_image). All metrics are the lower the better. We consistently observe LLaVA-1.5-7B-LORA instruction-tuned on both our CertainlyUncertain data + LLaVA data outperforming the base LLaVA-1.5-7B-LORA model, which is only trained on LLaVA data.
>
>     | Model | Mean Hallucination Ratio| hal_sents_ratio | hal_words_ratio | # hal_sents_per_image | # hal_words_per_image|
>     | -------- | -------- | -------- | -------- | -------- | -------- |
>     | LLaVA-1.5-7B-LORA (LLaVA Data Only)     | 0.316     | 0.316     | 0.342     | 1.595     | 30.41     |
>     | Ours+LLaVA Data     | **0.308**     | **0.308**    |**0.337**     | **1.55**     | **29.9**     |
>
> - **Results on MME**
>
>     We follow the original implementation of MME evluation and report the detailed scores for the sub-splits in MME below. Again, we consistently observe better performance from the model trained with ours+LLaVA data, with +8.634 and +11.072 improvements on Perception and Cognition scores, respectively.
>
>     | Model | Perception | Cognition |
>     | -------- | -------- | -------- |
>     | LLaVA-1.5-7B-LORA (LLaVA Data only)     | 1478.483     | 257.857     |
>     | Ours+LLaVA Data     | **1487.117**     | **268.929**    |
>
> We will try our best to include the full benchmarking results suggested by the reviewer to Table 6 in the final version.
>
> > Q3: Data Portion
>
> Thanks for pointing out, our design choice was originally to balance the data from different sources (*i.e.,* captions and images) to make the dataset more diverse. The challenge behind collecting more instances from captions is the lack of high-quality human-annotated descriptions which are highly compositional and include world knowledge, spatial relationships, visual settings, text rendering, and object attributes, like those in DOCCI. Therefore, we tried to leverage all training captions in DOCCI to create data for knowledge, complexity, temporal and ambiguity awareness categories. We also tried to avoid collecting multiple similar questions of the same type on the same caption to encourage more diverse questions. We believe we can further scale the data sourced from captions more significantly, with the recent release of PixMo-Cap [1], which we leave as future work.
>
> [1] Deitke, Matt, et al. "Molmo and pixmo: Open weights and open data for state-of-the-art multimodal models." arXiv preprint arXiv:2409.17146 (2024).
>
> > Q4: Concern of "Complexity Awareness"
>
> The goal of our benchmark is not simply to teach model to refuse, but to teach model to be aware of uncertainty, so that it can note its uncertainty or confidence when faced with unanswerable questions. We do observe models often express uncertainty in diverse forms, for example, provide a probable guess but explicitly note its confidence; bluntly refuse (e.g., “I don’t know” or “unanswerable”); asking a follow-up question; These observations motivated us to propose the two-stage evaluation pipeline which includes IDK-normalization where we use an LLM to judge whether either the prediction or ground truth (GT) is IDK and normalize the answer with IDK. It is important to note that by adopting an IDK normalization stage, our evaluation metric does not penalize the model when it answers with a probable guess but note the unanswerability of the question, like in the example of counting the balloons: "it is impossible to count exactly, but it’s probably 30”.

---

> > ### Comment · Reviewer_EAqJ · 2024-11-22
> >
> > Thank you for the efforts made in this rebuttal. Most of my concerns have been addressed. However, due to the limited model coverage in the proposed benchmark, I maintain my original rating.
> >
> > I strongly encourage the authors to complete the evaluation. I recommend utilizing the OpenCompass-VLMEvalKit* library to customize the benchmark and assess recent advanced models. I estimate that completing this will take only several hours.
> >
> > * https://github.com/open-compass/VLMEvalKit

---

> > > ### Author Response · Authors · 2024-11-25
> > > **Updated results with more model coverage**
> > >
> > > We sincerely thank the reviewer for the suggestion. We have updated Table 4 of the paper as suggested, with the following models:
> > > - Qwen2-VL-7B
> > > - Qwen2-VL-72B
> > > - LLaVA-OneVision-7B
> > > - LLaVA-OneVision-72B
> > > - InternVL2-8B
> > > - InternVL2-40B
> > > - InternVL2-76B
> > >
> > > We briefly summarize the results below.
> > >
> > > | Model | Total Confidence-weighted accuracy |
> > > | -------- | -------- |
> > > | Qwen-VL-Chat     | 13.45     |
> > > | LLaVA-1.6-7B     | 20.47     |
> > > | LLaVA-1.6-34B     | **32.47**     |
> > > | Qwen2-VL-7B     | 18.05     |
> > > | Qwen2-VL-72B     | 20.73     |
> > > | LLaVA-OV-7B     | 19.10     |
> > > | LLaVA-OV-72B     | 20.59     |
> > > | InternVL2-8B     | 9.04     |
> > > | InternVL2-40B     | 12.39     |
> > > | InternVL2-76B     | 13.63     |
> > >
> > > The results above again aligns with our previous observation: **Despite the big leap in performance from these models on standard benchmarks, they perform similarly to or even worse than the prior-art model LLaVA-1.6.**  Notably, LLaVA-OneVision has demonstrated stronger capabilities across single-image, multi-image and video scenarios over LLaVA-1.6, however the performance on CertainlyUncertain decreases even with larger model size (20.59 for LLaVA-OV-72B vs. 32.47 for LLaVA-1.6-34B).
> > >
> > > These results again highlight the contribution of our work, which extends existing benchmarks to focus on scenarios where clear and definitive answers are **not** available; and (2) provides a systematic data synthesize pipeline based on well-defined uncertainty taxonomy, so that VLMs can be trained to express uncertainty or acknowledge when they do not know the answer.
> > >
> > >
> > > We hope these updated results can address the reviewer's concern.

---

> ### Author Response · Authors · 2024-12-03
> **Updated results with more evaluations**
>
> We follow the reviewer's suggestion and report a more complete evaluation results on the four additional benchmarks, including SHR and HallusionBench for hallucination evaluation and MME, MMbench for standard evaluation. All metrics except the ones on SHR are the higher the better.
>
> | Model | SHR (mean hall. ratio) | HallusionBench (question acc.) | MME (preception+cognition) | MMBench-dev (circular evaluation acc.) |
> | -------- | -------- | -------- |-------- | -------- |
> | LLaVA-1.5-7B-LORA (the official checkpoint)     | 0.316 | 33.48     | 1736.34 | 74.91    |
> | Ours-only     | 0.398 | 31.71    | 1250.90 | 61.51    |
> | Ours+LLaVA Data     | **0.308** |   **34.46**     | **1756.05** | **75.60**  |
>
> The above results are consistent with our observation in the main paper, LLaVA-1.5-7B-LORA instruction-tuned on both our CertainlyUncertain data and LLaVA data outperforming the base LLaVA-1.5-7B-LORA model on hallucination-based benchmarks, while maintain or even improve on standard benchmarks.
>
> We hope our rebuttal fully addresses the reviewer's concern.

---

### Official Review · Reviewer_7h8B · 2024-11-03

**Soundness:** 3
**Presentation:** 4
**Contribution:** 4
**Rating:** 6
**Confidence:** 4

**Summary:**

The work aims to address the uncertainty in vision-language models (VLMs). They first present a taxonomy of uncertainty resulting from a lack of information or inherent unpredictability. They then synthesize a benchmark dataset named CERTAINLYUNCERTAIN, and introduce a new metric, confidence-weighted accuracy. Besides, they further demonstrate that supervised fine-tuning with CERTAINLYUNCERTAIN enhances the performance of VLMs, and reduces the calibration error.

**Strengths:**

-	The work aims to address the important problem of uncertainty in VLMs. An interesting and valuable benchmark is proposed.
-	The work shows clear experimental evidence that demonstrates the improvements from fine-tuning with the proposed dataset.
-	The experimental results provide clear evidence of the benefits of the proposed dataset. The fine-tuned model with CERTAINLYUNCERTAIN shows performance gains and improved calibration.
-	The paper is well-written with clear presentation.

**Weaknesses:**

-	While the process of deriving contrastive instances in CERTAINLYUNCERTAIN from image and caption sources is clearly described, it is unclear how those instances are classified into the predefined uncertainty types (i.e., Epistemic, Aleatoric).
-	The dataset shows bias: a high proportion of QA pairs is categorized as “extraneous awareness”.
-	It would be helpful to know the human performance on this benchmark for comparison.

**Questions:**

See weakness

---

> ### Author Response · Authors · 2024-11-21
> **Response to Reviewer 7h8B**
>
> Thank you for the encouraging review. Please find our response below.
>
> > Q1:  It is unclear how contrastive instances in CERTAINLYUNCERTAIN from image and caption sources are classified into the predefined uncertainty types (i.e., Epistemic, Aleatoric).
>
> We collect instances for knowledge, complexity, temporal and ambiguity awareness categories from caption sources (L185), which do not require image perturbation, but can be more easily inferred from comprehensive human-annotated captions. While sourcing from images are solely for extraneous category, which we leverage image perturbation to test whether the model can *identify and disregard elements within an image that are not relevant to the question at hand*.
>
>
> > Q2: Data imbalance
>
> Thanks for pointing out, our design choice was originally to balance the data from different sources (*i.e.,* captions and images) to make the dataset more diverse. The challenge behind collecting more instances from captions is the lack of high-quality human-annotated descriptions which are highly compositional and include world knowledge, spatial relationships, visual settings, text rendering, and object attributes, like those in DOCCI. Therefore, we tried to leverage all training captions in DOCCI to create data for knowledge, complexity, temporal and ambiguity awareness categories. We also tried to avoid collecting multiple similar questions of the same type on the same caption to encourage more diverse questions. We believe we can further scale the data sourced from captions more significantly, with the recent release of PixMo-Cap [1], which we leave as future work.
>
> [1] Deitke, Matt, et al. "Molmo and pixmo: Open weights and open data for state-of-the-art multimodal models." arXiv preprint arXiv:2409.17146 (2024).
>
> > Q3: Human performance
>
> Thanks for the suggestion. We conduct a small scale human evaluation on the 100 randomly sampled questions. As it is impossible to truly measure human confidence on their predictions, we report LAVE Acc. instead of confidence-weighted accuracy. Overall, human achieves much higher score of 94% than models, with close to perfect score of 98% on unanswerable questions (with only 1 wrong answer, 49 correct answers), while slightly lower (90.0%) on answerable questions.
>
> In this small scale evaluation, we found that humans tend to provide brief answers to answerable questions. For example, when asked about the color of a mostly black clothing with some patches of white, humans may provide answers as `black`, while the ground truth answer is `mostly black with some patches of white`. In this case, the answer `black` only receives partial scores. Another interesting observation is that humans tend to bluntly refuse more than models (such as GPT4V), as it takes less efforts to say "I don't know" than providing a probable guess or engaging more actively by asking for follow-up questions.

---

> > ### Author Response · Authors · 2024-12-03
> >
> > Dear Reviewer 7h8B,
> >
> > Thank you again for your constructive feedback! We believe all your concerns and suggestions are addressed and incorporated now.
> >
> > Therefore, we kindly wonder if you can increase your rating to reflect it and support our work. We’re also happy to hear any additional suggestions to further improve the paper. Thank you for your time and review.

---

### Official Review · Reviewer_FPfF · 2024-11-04

**Soundness:** 3
**Presentation:** 2
**Contribution:** 3
**Rating:** 6
**Confidence:** 3

**Summary:**

The authors introduce a benchmark called CertainlyUncertain, consisting of 178K VQA pairs. This dataset consists of both answerable and unanswerable questions, where the correct answer to the latter is "I don't know". They introduce a taxonomy of five types of uncertainty, and generate questions via two methods: either caption-based prompting with GPT4 or inpainting of salient image regions for answerable questions. They show that existing VLMs do not perform well on the task; however, fine-tuning improves performance and also improves performance on other datasets that are refusal-based or test hallucinations. They also introduce a metric for measuring confidence-aware accuracy.

**Strengths:**

The paper focuses on an important capability of ML models, the capacity to express uncertainty when appropriate. They introduce a large and diverse dataset to explicitly assess uncertainty of VLMs in a VQA setting. Different sources of uncertainty are nicely categorized into a taxonomy, and questions are collected according to this taxonomy. Moreover the same image contains both answerable and unanswerable questions in their dataset, providing a nice contrastive setup.

They benchmark SoTA VLMs against their dataset, and show that fine-tuning these models on their dataset leads to better performance on both their dataset and on other relevant datasets (refusal-based, hallucinations). Experiments seem reasonably thorough across different datasets and metrics are broken down for different types of uncertainty.

**Weaknesses:**

Dataset quality is a concern I have. As noted by the authors, this is a concern with automatically-generated datasets. They note that 20% of the samples were filtered out on a quality check, which is a reasonably high number. Was a similar filtration process applied to the training set for the extraneous questions, or were those not filtered? Also were the questions generated from image captions also quality-checked? I would like to understand more about the 93% number they quote in Ln 239.

Some qualitative analysis in the results would be helpful, e.g. showing how training on the dataset causes the model to perform better on questions that involve uncertainty, compared to the base model. The lack of qualitative examples makes the results difficult to read and it would be good to add such examples.

Clarity of the paper could be improved significantly. For example, certain terminology such as the LAVE method is introduced without proper explanation, although it is important to understand how this works. More explanation on the hallucination-based benchmarks would be helpful. The experiments section is quite dense with large tables -- as mentioned above, some qualitative results would be helpful. It is unclear whether the entirety of Table 4 needs to be presented or whether some numbers can be moved to the Appendix.

I think the paper would benefit from a background section, which explains some key concepts such as VLMs and SoTA models, common metrics used for evaluating their uncertainty capabilities, fine-tuning strategies that you use in the experiments section, anything else that is important. Related work is dense and would also benefit from better explaining some of the previous relevant datasets and work on multimodal uncertainty or refusal that has been explored. These could be explained more clearly.

Lns 195-196 vs. 198-200: It is not clear whether the questions are from the original dataset and only the images are perturbed, or whether the questions are generated from GPT4-V after the image perturbation has been applied.

**Questions:**

Most of my questions/comments are elaborated on in the above section. Please take note of my comments on dataset quality and clarity/reorganization.

---

> ### Author Response · Authors · 2024-11-21
> **Response to Reviewer FPfF**
>
> Thank you for the detailed review. Please find our response below.
>
> > Q1: Data Quality
>
> As we have stated in Appendix Section C, using synthetic model-generated data for training has become a de facto approach in the literature, where previous works [1,2,3] have consistently observed great performance improvement even with noisy labels on synthetic model-generated data. `This is also consistent with our observation in Table 6` (previously Table 5) and `the additional experimental results on SHR and MME in the response to Reviewer EAqJ`, where training with CertainlyUncertain has demonstrated strong performance across multiple benchmarks. Therefore, `we focus our manual efforts with humans on quality check and filtering for all testing questions of CertainlyUncertain`.
>
> The detailed results of our human filtering process on each sub-category of the testing split have been shown in Appendix Table 6. For extraneous category, we directly discard the invalid ones. For the other categories which are generated from DOCCI image captions, we retain the first 5K samples verified by human on DOCCI testing images to build the remaining testing splits (L239 - 240).
>
> We agree with the reviewer that data quality of the training split is important, however, performing large-scale quality check on training questions is much more labor-intensive.  For training splits other than extraneous, we expect similar quality to the testing questions (*i.e.* > 93%). For extraneous training split, 53K out of 76K training questions are based on groundtruth scene graph annotations in GQA (L202 - 206). Though these questions have their limitations being not very natural, as they are rule-based, they are more reliable than GPT-generated QAs.
>
> [1] Liu, Haotian, et al. "Visual instruction tuning." Advances in neural information processing systems 36 (2024).
>
> [2] Zhu, Deyao, et al. "Minigpt-4: Enhancing vision-language understanding with advanced large language models." arXiv preprint arXiv:2304.10592 (2023)
>
> [3] Chen, Lin, et al. "Sharegpt4v: Improving large multi-modal models with better captions." arXiv preprint arXiv:2311.12793 (2023).
>
> > Q2: Qualitative Analysis
>
> Due to the space limit for the main paper, we have provided qualitative analysis in Figure 11 of the Appendix, with accompanying discussion in Appendix B text. Figure 11 not only qualitatively shows `how training on the dataset causes the model to perform better on questions that involve uncertainty, compared to the base model`, but also demonstrate `how the proposed confidence-weighted accuracy gives less score for a correct answer with lower confidence, and penalizes more for an incorrect answer with higher confidence.`
>
> We appreciate the reviewer's recommendation of moving part of Table 4 (the new Table 5) to appendix, while moving the qualitative analysis to the main text, we will consider it for the final version.
>
> > Q3: Paper presentation
>
> Thanks for the suggestion. We have added more details to clarify on LAVE metric, hallucination-based benchmarks, finetuning strategies in the corresponding sections (highlighted in blue).
>
> To the best of our knowledge, there are no common metrics specifically used for evaluating uncertainty capabilities of VLMs, we have summarized the commonly used ones for evaluating VLMs in Related Works. In addition to that, we added the new Table 3 to provide a holistic view of evaluation metrics in this space, and to compare against our proposed metric. Table 2 has provided a bird-eye view of previous relevant datasets and work on multimodal uncertainty or refusal.
>
> We will try our best to refine the related work and add more details about the background of VLMs for the final version.
>
> > Q4: Lns 195-196 vs. 198-200
>
> Sorry about the confusion. Lns 195-196 is about the general pipeline, which is shared among VQAv2 and GQA images.
>
> The questions based on VQAv2 images are generated from GPT4V (L198-200). We first leverage the original VQAv2 questions to find salient objects that are question-worthy, then perform perturbation and send to GPT4V to generate questions that are answerable for the original images, while unanswerable for the perturbed images. This is because that we empirically found that most of the VQAv2 questions are not directly usable. For example, there are questions like "What is the person wearing?" for images with multiple people, which creates challenges to the Grounded-SAM model for masking all of them.
>
> Thanks to the rule-based questions and dense scene graph annotations in GQA, we can directly take the questions from the original GQA dataset, and leverage groundtruth annotations for both saliency identification and grounding (*i.e.,* extracting the object bounding box for SAM model to predict the mask). For instance, if asking a similar question like "What is the person wearing?" for image with multiple people, we directly mask out all person regions in the image.
>
> We have modified the corresponding text to clarify this.

---

> ### Comment · Reviewer_FPfF · 2024-11-26
>
> I thank the reviewers for their detailed response to my comments and the comments of the other reviewers. I appreciate the clarification on dataset quality, and it is good to know that all the test questions have undergone manual verification. The modifications to the main text are also appreciated, and improve the clarity of the paper. For qualitative examples, I would suggest providing some examples in the main paper (space allowing) where the fine-tuned model succeeds and the base model fails, and analyzing what kinds of questions and uncertainty categories show a particular improvement in the fine-tuned model. Moreover some analysis of model failures on the test set and other datasets in Table 6 would be useful to understand limitations.
>
> I appreciate the response to another reviewer's comments about further model and dataset evaluations. It is good to see evaluation of more SoTA VLMs. For datasets, improvement is good on refusal-based datasets. I also believe that evaluation of more external datasets is important, such as hallucination-based datasets as suggested by reviewer EAqJ. I appreciate the evaluation on some benchmarks in the authors' response to that reviewer, where the results are promising. I would ask the authors to expand on this evaluation with a few more VLMs and potentially other benchmarks suggested by the reviewer, and add those to Table 6. In the absence currently of those results in the paper, I will keep my current rating, which is already positive. That said, the authors have suggested that they can add these results which would be helpful and look promising. Thus, in light of the strengths of the paper and the adequate response to my concerns, I am inclined to recommend acceptance of the paper.

---

> ### Author Response · Authors · 2024-11-27
>
> We appreciate the reviewer's response to our rebuttal. We are actively working on adding the other benchmarks suggested by reviewer EAqJ to Table 6. In the meantime, we would like to better understand the reviewer's suggestions about including additional VLMs for Table 6.
>
> We mainly based Table 6 experiments on LLaVA-1.5, due to its open-source training code, stage 1 pre-training weights and training data for instruction tuning. LLaVA-1.6 did not open-source their training data, while LLaVA-OneVision requires training on video data, which is resource-demanding, and out of the scope of this paper. Other models like Qwen-VL, Qwen2-VL, InternVL2 only open-source their final model (refer to [this link](https://molmo.allenai.org/blog) for a detailed comparison). Another candidate could be Molmo, however their training data and code is not yet publically available.  Though continued SFT with our data is feasible, similar to how we experimented with Qwen-VL-Chat. It is a less ideal setting when the original instruction tuning data for Qwen-VL-Chat is not available, which is why we added LLaVA-Data for the continued SFT.
>
> We would love to hear about what other VLMs the reviewer have in mind, so that we can further strengthen the paper.

---

> ### Comment · Reviewer_FPfF · 2024-11-27
>
> Thanks for your reply, and I appreciate the explanation. I think I would be happy with experiments on LLaVA-1.5 as you have done in Table 6 -- comparing LLaVA-data instruct tuning, instruct tuning on your dataset, and instruct tuning on LLaVA-data+yours, and showing the results across these external benchmark datasets. What I was suggesting was similar to the Qwen-VL-Chat experiments in Table 6, i.e. SFT on your dataset and also yours+LLaVA-data and then reporting performance on the benchmarks. However, I think I would be convinced by experiments on LLaVA-1.5 which should be sufficiently convincing, and as you have explained there are advantages to LLaVA-1.5 for instruct-tuning. Therefore I revise my statement and say that adding other VLMs is not necessary in my view for these experiments. I would just suggest augmenting Table 6 for LLAVA-1.5 with results on other benchmarks.

---

> ### Author Response · Authors · 2024-12-03
>
> We follow the reviewer's suggestion and report a more complete evaluation results on the four additional benchmarks suggested by reviewer EAqJ, including SHR and HallusionBench for hallucination evaluation and MME, MMbench for standard evaluation. All metrics except the ones on SHR are the higher the better.
>
> | Model | SHR (mean hall. ratio) | HallusionBench (question acc.) | MME (preception+cognition) | MMBench-dev (circular evaluation acc.) |
> | -------- | -------- | -------- |-------- | -------- |
> | LLaVA-1.5-7B-LORA (the official checkpoint)     | 0.316 | 33.48     | 1736.34 | 74.91    |
> | Ours-only     | 0.398 | 31.71    | 1250.90 | 61.51    |
> | Ours+LLaVA Data     | **0.308** |   **34.46**     | **1756.05** | **75.60**  |
>
> The above results are consistent with our observation in the main paper, LLaVA-1.5-7B-LORA instruction-tuned on both our CertainlyUncertain data and LLaVA data outperforming the base LLaVA-1.5-7B-LORA model on hallucination-based benchmarks, while maintain or even improve on standard benchmarks.
>
> We hope our rebuttal fully addresses the reviewer's concern.

---

### Author Response · Authors · 2024-11-21
**General Response**

We sincerely thank all the reviewers for their time and constructive feedback. We are encouraged by the reviewers' recognition that:
- Our paper is **well-written with clear presentation** (Reviewer `7h8B`), **focuses on an important problem of uncertainty** in VLMs (Reviewer `FPfF`,  `EAqJ`, `7h8B`).
- Our proposed uncertainty taxonomy is **rigorous, well designed** (Reviewer `EAqJ`), and **nicely categorized** (Reviewer `FPfF`).
- Our **large and diverse** (Reviewer `FPfF`) dataset is **interesting, valuable** (Reviewer `7h8B`), **novel and a commendable effort** (Reviewer `EAqJ`).
- Our proposed confidence-weighted accuracy is **innovative**, offers **a comprehensive evaluation** by factoring in both the correctness and confidence of predictions (Reviewer `EAqJ`).
- Our experiments are **reasonably thorough** (Reviewer `FPfF`),  **comprehensive** (Reviewer `EAqJ`), **provide clear evidence of the benefits of the proposed dataset** (Reviewer `7h8B`)

Please find our detailed responses to specific questions from the reviewers in each rebuttal.

---

### Meta-Review · Area_Chair_BYMQ · 2024-12-22

**Metareview:**

Paper introduces a novel VQA benchmark for VLMs consisting of answerable and unanswerable questions. The latter is the unique property of the proposed dataset, where the expected answer includes assertion of the model not being able to answer and the reason. The paper also introduces a taxonomy of different uncertainties and programmatically generated questions for the benchmark. Benchmarking of existing VLMs on the datasets illustrates that quantification of uncertainty is substantially lacking.

The paper was reviewed by three expert reviewers, which gave it 3 x marginally above the acceptance threshold. The reviewer concerns centered around (1) quality of the dataset [FPfF], (2) lacking exposition [FPfF, 7h8B], (3) lack of human performance baseline [7h8B], (4) potential dataset bias [7h8B, EAqJ], and (5) limited evaluated model coverage [EAqJ]. Authors have done an excellent job addressing these concerns in the rebuttal; with reviewer [FPfF] acknowledging that his and other reviewer's concerns were adequately addressed. Unfortunately the remaining two reviewers did not participate in discussion.

AC has read the review, rebuttal and discussion that followed, as well as the paper itself. AC believes the task is interesting and the dataset is important for the community (deviating from other task-specific benchmarks). While initial submission did have significant issues raised by reviewers, these were largely addressed during the rebuttal. Specifically, given that the test split has been manually verified, addresses issue (1); exposition has been improved some (2); human baseline provided (3), and number of models evaluated greatly improved (5). While the broader evaluation of models would still be helpful, on the balance AC believes the work is valuable and important and will be a net benefit to the community. Therefore AC is recommending Acceptance.

Authors are encouraged to incorporate results and arguments made during the rebuttal into the main paper, including additional evaluations of SoTA models made in response to [EAqJ].

**Additional Comments On Reviewer Discussion:**

During discussion, reviewer [FPfF] acknowledged that his and other reviewer's concerns were adequately addressed. [FPfF] comments that rebuttal provided "adequate response to [...] concerns, I [he/she is] inclined to recommend acceptance of the paper." Unfortunately, the remaining two reviewers did not participate in discussion. AC has validated, however, that their concerns were adequately addressed by the authors, resulting in the overall recommendation above.

---

### Decision · Program_Chairs · 2025-01-22

Accept (Poster)